# Fertilizers and Fertilization Strategies Mitigating Soil Factors Constraining Efficiency of Nitrogen in Plant Production

**DOI:** 10.3390/plants11141855

**Published:** 2022-07-15

**Authors:** Przemysław Barłóg, Witold Grzebisz, Remigiusz Łukowiak

**Affiliations:** Department of Agricultural Chemistry and Environmental Biogeochemistry, Poznan University of Life Sciences, Wojska Polskiego 71F, 60-625 Poznan, Poland; witold.grzebisz@up.poznan.pl (W.G.); remigiusz.lukowiak@up.poznan.pl (R.Ł.)

**Keywords:** crop growth rate, fertilizer market, nitrogen use efficiency, nitrogen gap, nutrient uptake, partial factor productivity, root architecture

## Abstract

Fertilizer Use Efficiency (FUE) is a measure of the potential of an applied fertilizer to increase its impact on the uptake and utilization of nitrogen (N) present in the soil/plant system. The productivity of N depends on the supply of those nutrients in a well-defined stage of yield formation that are decisive for its uptake and utilization. Traditionally, plant nutritional status is evaluated by using chemical methods. However, nowadays, to correct fertilizer doses, the absorption and reflection of solar radiation is used. Fertilization efficiency can be increased not only by adjusting the fertilizer dose to the plant’s requirements, but also by removing all of the soil factors that constrain nutrient uptake and their transport from soil to root surface. Among them, soil compaction and pH are relatively easy to correct. The goal of new the formulas of N fertilizers is to increase the availability of N by synchronization of its release with the plant demand. The aim of non-nitrogenous fertilizers is to increase the availability of nutrients that control the effectiveness of N present in the soil/plant system. A wide range of actions is required to reduce the amount of N which can pollute ecosystems adjacent to fields.

## 1. Fertilizer Use Efficiency—A Real Farming Practice

### 1.1. Nitrogen Gap and the Maximum Attainable Yield

A farmer needs to recognize production boundaries in order to develop an effective production program for each of the crops grown on the farm. The key to the sound management of production processes is a knowledge of the maximum yield that can be achieved in a production area with a well-defined climate and soils. The actual yield (Y_a_) of a currently cultivated crop may be simply presented as the difference between the maximum attainable yield (Y_attmax_) and the yield gap (YG). The relationship between these terms may be expressed as the formula:(1)$$Y_{a}=Y_{\mathrm{attmax}}-\mathrm{YG}$$

Y_a_ is a real, harvested yield in the current growing season under actual environmental, agronomic and management practice on the farm. To define the Y_attmax_ of this crop, two conditions must be fulfilled. The first concerns a strictly defined climatic area and the dominating, i.e., standard, weather conditions [1,2]. The second necessary condition is the level of soil fertility, agronomic conditions and management of the production processes on the farm. These factors modify the Y_attmax_ of the grown crop [3,4]. All of these factors must be oriented towards optimizing the supply of nutrients to that particular crop only [5]. The YG is a measure of the ineffectiveness of production factors, in fact expressed in the ineffectiveness of fertilizer nitrogen (N_f_), or available N present in the soil/plant system during the growing season of the currently grown crop [6]. The basic and at the same time simplest method for calculating both components of the Y_a_ formula is to use the efficiency index of N_f_ known as the Partial Factor Productivity of Fertilizer N, PFP_Nf_ [7]. Considering both the yield and the environmental aspects of the on-farm production process, the farmer’s goal should not be to determine the YG, but rather the ineffectiveness of the applied N_f_. The quantitative expression of N inefficiency is the nitrogen gap (NG) [8]. In fact, two sets of data are needed to determine both the YG and the NG, i.e., (i) the actual yield harvested by the farmer, and (ii) the amount of applied N_f_. The calculation procedure consists of a set of formulas:

Partial Factor Productivity of N_f_:(2)$${\mathrm{PFP}}_{\mathrm{Nf}}=\frac{\mathrm{Ya}}{N_{f}}\left({\mathrm{kg}\;\mathrm{kg}}^{-1}\;N_{f}\right)$$

Attainable, maximum yield: (3)$$Y_{\mathrm{attmax}}={\mathrm{cPFP}}_{\mathrm{Nf}}\times N_{f}\left(t\;\mathrm{or}\;\mathrm{kg}\;{\mathrm{ha}}^{-1}\right)$$

Yield Gap:(4)$$\mathrm{YG}={\;Y}_{\mathrm{attmax}}-{\;Y}_{a}\;\left({t\;\mathrm{ha}}^{-1}\right)$$

Nitrogen Gap: (5)$$\;\;\;\;\;\;\;\;\;\;\;\mathrm{NG}=\frac{\mathrm{YG}}{{\mathrm{cPFP}}_{\mathrm{Nf}}}\left({\mathrm{kg}\;N\;\mathrm{ha}}^{-1}\right)$$
where: PFP_Nf_—partial factor productivity of N_f_, kg grain/seeds, tubers etc. per kg N_f_; Y_a_—actual yield of a currently grown crop, t ha^−1^; N_f_—the amount of applied fertilizer N, kg ha^−1^; Y_attmax_—the maximum attainable yield, t ha^−1^; cPFP_Nf_—the average of the third quartile (Q3) of the set of PFP_Nf_ indices arranged in ascending order, kg grain/seeds, tubers etc. per kg N_f_; YG—yield gap, t ha^−1^_;_ NG—nitrogen gap, kg ha^−1^ of N.

The NG calculation is important for the farmer for at least three areas of his production activity: (i) the determination of Y_attmax_, which determines not only the maximum yield for the production area, but also determines the potential requirements of the cultivated crop for N; (ii) the identification of hotspots in N management for a given crop, including an inadequate supply of nutrients other than N; (iii) the set of actions needed to improve the level of soil fertility for a given crop. 

The data on NG is used to construct a diagram of the impact of the NG change on trends in actual and maximum yields (Figure 1). The target of the NG construction is to find the maximum attainable yield (Y_attmax_) for the geographical area of the farm operation. The Y_attmax_ value is determined by the intersection of Y_max_ and Y_a_ linear regression models. In this specific case, representing 16 fields located in a small region of central-western Poland, the weather and soil conditions are stable. Y_attmax_ for winter wheat reached 7.99 t ha^−1^. Moreover, both Y_a_ and Y_attmax_ showed significant variability in the amount of *notworkable* N_f_ during the growing season. The course of both models indicates a surplus of N_f_ on fields No. 13 and 10 as the main reason for its lower use efficiency. The maximum YG on field No. 13 reached 3.729 t ha^−1^, i.e., it constituted 47% of the actual yield. The diagnostic goal of the NG diagram construction is to identify the key factors responsible for YG appearance as a result of N_f_ inefficiency. The ranges for the evaluation of the effect of any production factor were constructed using a clear scale: low, medium, high, which were in special cases underlined by “very”. The use of this scale to assess the production effect of N_f_ is shown in Appendix A.

### 1.2. Fertilizer Use Efficiency—FUE 

The term Fertilizer Use Efficiency—FUE is not new. It has been widely used for decades but has become widespread recently thanks to the use of the FUE indexes to assess the global productivity of NPK fertilizers [7,9]. The productivity of nutrients applied in fertilizers can be estimated by the same formula as shown in Equation (2) for fertilizer N. Another methodological way for FUE determination is to use a set of indices used in field experiments such as Apparent Nutrient Efficiency (ANeE) and/or Apparent Nutrient Recovery (ANuR):(6)$$\mathrm{ANuE}=\frac{Y_{f}-{\;Y}_{c}\;}{N_{r}}$$
(7)$$\mathrm{ANuR}=\frac{{\;\mathrm{Nu}}_{f}-{\mathrm{Nu}}_{c}}{N_{r}}$$
where: ANuE—Apparent Nutrient Efficiency, kg yield kg^−1^ nutrient applied; ANuR—Apparent Nutrient Recovery, %; Y_f_, Y_c_—yield on a plot with and without fertilizer, t or kg ha^−1^; N_r_—the rate of a nutrient applied as fertilizer, kg or g ha^−1^; Nu_f_, Nu_c_ —the uptake of a tested nutrient on a plot with and without fertilizer, kg, g ha^−1^. 

The recorded values of ANuE and ANuR usually show a decreasing trend, with an increase in the rate of the nutrient applied as fertilizer, which is satisfactory for the researcher. Moreover, the values obtained have a tendency opposite to the soil fertility indexes for a given nutrient [7]. It simply means that FUE is highly dependent on the current soil fertility level, which the farmer needs to know. However, the main disadvantage of these two indices is that the farmer does not have a control plot to assess the actual nutrient productivity in the applied fertilizers. The values of the ANuR indices, evaluated on the global scale, are low and amount to 40–65% for N, 15–25% for P, and 30–50% for K used in fertilizers [9]. At this point it is necessary to pose the question, what is the main source of nutrients for the currently grown crop? 

The productivity of nutrients taken up by the crop during one growing season can also be estimated by the partial nutrient balance (PNB) method:(8)$$\mathrm{NuE}=\frac{{\mathrm{Nu}}_{t}}{{\mathrm{Nu}}_{f}}\times 100\%$$
where: NuE—Nutrient uptake Efficiency, kg kg^−1^; Nu_t_—the uptake of a tested nutrient, kg or g ha^−1^; Nu_f_—the rate of a nutrient applied as fertilizer, kg, g ha^−1^. 

The efficiency of N, P, and K using this method show much higher values or even a surplus of nutrients [10]. The low efficiency of nutrients using the differential methods, but high yield indirectly indicates that the main source of nutrients for crops grown in one growing season is soil [11].

The main problem is the assessment of the production role of nitrogen, which plants take in in two distinct inorganic forms, i.e., as nitrate (NO_3_^−1^) and ammonium (NH_4_^+^) [12]. Nitrates affect plant growth in many ways, inducing plant morphology, physiology through hormones and finally metabolism through their influence on the production of organic acids [13,14,15]. Plants fed with nitrates, compared to ammonium, show a high growth rate, which results in higher yields [16]. The above-identified aspects of the impact of N on plants are fully supported by field experiments and agricultural practice [17,18]. As shown in Figure 2, the yields of winter wheat grown on the control plot (non-fertilized) and on the plots fertilized with K, P in the same way since 1957, did not show large differences. The average yield for these three objects of 4.38 ± 0.14 t ha^−1^, can be considered as high. The primary reason for such a high yield, despite the lack of N fertilization, was alfalfa as a forecrop. The use of 90 kg N ha^−1^ increased the yield by 1.94 t ha^−1^. The same level of yields was also recorded for the NP and NK plots. The lack of response to the P or K application clearly emphasizes the importance of these two nutrients for plant growth and yield. This conclusion was fully confirmed by the yield achieved on the NPK plot. Even more important is the fact that N use efficiency (NUE) increased by 10–13%, compared to incomplete fertilization treatments. The observed interaction was even more important for P use efficiency (PUE), which in the NPK plot increased by 9% and by 73% compared to NP and P treatments, respectively. The same trend was observed for potassium. The importance of the N × PK interaction on the productivity of N_f_ is observed for all crops, regardless of the world region [17,19,20]. The complex effect of N on plant growth and yielding clearly indicates the superior function of N in crop production. It can, therefore, be concluded that the production efficiency of nutrients, applied as mineral fertilizers, can be mainly evaluated through their impact on NUE. Thus, the search for indicators of productivity or efficiency for other nutrients is pointless. This is well presented in the analysis of the causes of the NG (Appendix A).

### 1.3. Factors Affecting Fertilizer Use Efficiency 

Fertilizer use efficiency is the result of a series of interactions between plant genotype and environment, including both abiotic and biotic factors. Full recognition of these factors is the basis for proper fertilization of plants in farming practice, aimed at maximizing the FUE values. The soil is both the growth environment for plants and their main reservoir of water and nutrients. Hence, the impact of soil factors on nutrient uptake and FUE should be considered at the level of several groups of phenomena and processes (Figure 3). 

In the first group (A) all of the factors, both abiotic and biotic, that lead to the release of nutrients from their solid phase in the soil to their solution phase should be analyzed. The next group of factors (B) is concerned with the processes of transporting nutrients from the soil to the root surface. The third group (C) of factors influencing FUE concerns plant responses manifested by changes in architecture and root growth rate. This group of factors, also related to plant activity, should consider the composition of the root exudates in the plant root—mycorrhizal system. For the assessment of the effectiveness of fertilizer application, the processes taking place in the plant itself, related to transport, assimilation in the aboveground mass (D), as well as remobilization of components and their transfer from the vegetative parts to the generative crop (E), are also important.

## 2. Factors Affecting Nutrient Uptake

### 2.1. Plant Growth and Nutrient Requirement 

A major challenge for the farmer is to synchronize the crop plant requirement for nutrients with their supply from both soil and applied fertilizers. The term synchronization refers to the amount of a nutrient that must be taken up by the crop at a certain stage of its growth as a prerequisite for a development of yield components. The expected degree of a given yield component formation depends on the growth rate of the crop, which in turn depends on the supply of N. For example, the critical stage of yield formation by winter oilseed rape (WOSR) reveals itself at the phase of inflorescence development (BBCH 50–59; coding system of growth stages, abbreviation in German: Biologische Bundesanstalt, Bundessortenamt und CHemische Industrie [21]). As shown in Figure 4, WOSR fertilized with N as ammonium nitrate (AN) in two equal rates of 80 kg N ha^−1^ applied at BBCH 22 (spring restart of WOSR) and BBCH 30/31 reached the maximum growth rate (CGR_max_, 20.4 g m^−2^ day^−1^) at full flowering. This value was the prerequisite of both the highest yield and the lowest, in its year-to-year variation (coefficient of variation, CV of 5.6%). 

For comparison, plants fertilized with the same N dose, but applied as calcium ammonium-nitrate (CAN), yielded on average at the same level, but showed much higher year-to-year variability. The main reason was a slightly lower CGR_max_ (19.1 g m^−2^ day^−1^), resulting in a higher CV (16%). Moreover, plants fertilized with AN reached the maximum N accumulation at the full flowering stage (BBCH 65), while those fertilized with CAN much later, i.e., at the beginning of pod growth (BBCH 71). The observed delay was due to the excessive growth of secondary branches, which is not always coordinated with higher yield [23]. The lower yield on the N control plot was mainly due to a significantly lower rate of dry matter accumulation, which resulted in a much worse status of the yield components at maturity. The existing relationship between nutrient uptake by a plant and its growth rate can be summarized by the equation [24]:(9)$$U_{\min}={\;C}_{c}\times \frac{\mathrm{dW}}{\mathrm{dt}}\times \frac{1}{w}\times \frac{W}{2\pi \mathrm{rL}}\;\;\mathrm{or}\;\;U_{\min}={\;C}_{c}\times \mathrm{RGR}\;\times \frac{W}{2\pi \mathrm{rL}}$$
where: U_min_—minimum uptake of a nutrient for the maximum rate of plant growth, g or kg plant^−1^ or unit area; C_c_—critical concentration of a nutrient in a plant, g, mg kg^−1^ DW; W—aboveground biomass of a plant, g or kg DW; r—root diameter, mm or cm; L—root length, cm or m; 2πrL—root surface area, mm^2^ or cm^2^ or m^−2^; $\frac{\mathrm{dW}}{\mathrm{dt}}\times \frac{1}{W}$—the relative growth rate of a plant, RGR, g g^−1^ t^−1^; t—time: day or year.

This equation clearly shows that the minimum amount of a given nutrient taken up by a plant over a specific period of time is necessary to maintain its critical concentration in plant tissues, determining the plant’s optimum growth rate. In the numerator of the equation, apart from the nutrient concentration, is the plant biomass, determined by two factors, i.e., the period duration (t—time) and the root surface area, as the denominator. 

The first challenge for the farmer in exploiting the yielding potential of the grown crop is to recognize the critical stage(s) of yield formation, or more precisely, the formation of yield components. Plant crop development is usually described on a 100-point scale (stages), divided into 10 phases [25]. Farmers need to know this scale to control the development of yield components. However, its use by the farmer for precise fertilization requires identifying those stages, which are crucial for the development of the main yield component. The degree of its development is closely related to the crop biomass, which is described by the sigmoid crop growth model [26]. The accumulation of crop biomass during the growing season, based on this model, shows variable growth rates in different phases, which fits with exponential, linear, quadratic or linear-plateau regression models (Figure 5). This trait can then be used to determine the three growth mega-phases of crops [8,27]:Exponential → Crop Foundation Period—CFP;Linear → Yield Formation Period—YFP;Quadratic or linear plateau → Yield Realization Period—YRP.

The first mega-phase refers to all crops, but the last one only to seed plants. The intersection points of CFP and YFP as the first pair, and YFP and YRP as the second, termed as cardinal knots (CKs), are two crucial points of the crop yield development [8]. CK1 is the change point at which the crop changes its rate of dry matter accumulation from the exponential to the linear model [26]. CKs are used by farmers as diagnostic steps to assess the crop nutritional status. CK1 is a crucial point at which to correct the nutritional status of all crop plants, regardless of the species [28]. In the case of cereals, CK1 refers to the borderline of tillering and the beginning of the stem elongation phase (BBCH 29–31). For dicots, this cardinal knot is related to the rosette stage. A classic example is winter oilseed rape ([25]; Figure 6). The critical nutrient concentration specified at CK1 is important, mainly for correcting the N status of the currently grown plant. For most crops, CK1 is the date of the maximum relative growth rate (RGR) of the crop. A classic example is maize. As shown in Figure 7, maize reached the maximum RGR on the 48th day after sowing (BBCH 15 to 17) and then its value decreased with increasing maize biomass. This particular period of maize growth is associated with the appearance of inflorescences [29,30]. Thus, the date when the plant reaches its maximum RGR defines the first cardinal phase of yield formation by the crop, i.e., CK1.

Moreover, as shown in Figure 7, the zinc (Zn) foliar treated maize maintained, more strictly extended the duration of the RGR peak. As a consequence of the prolonged biomass growth at BBCH 15–17, a second RGR peak, but much smaller, appeared during flowering. The yield increases due to zinc application before the CK1 resulted in a yield higher by 1.49 t ha^−1^. The partial factor of N productivity (PFP_Nf_) increased from 66.7 to 79.3 kg grain per kg of N_f_. The direct reason for the yield increase was the uptake of an N increase of 46.4 kg ha^−1^ [31]. The given example clearly indicates that the use of macronutrient fertilizers requires the precise diagnosis of the critical phase (s) of yield formation by the crop.

The second cardinal phase (CK2) is very well-defined for seed crops. This stage proceeds the date of flowering (Figure 8). For some crops, their nutritional status at CK2 can be used to forecast the yield. A classic example is maize. The nutrient content at this stage in the cob leaf is used to indicate the nutritional status of maize and delivers a highly reliable yield prognosis [32,33]. The same rule is observed for winter oilseed rape. The content of nutrients in leaves at flowering can be used to forecast the seed yield [34]. This relationship explains the opinion of Schulte auf’m Erley et al. [21] on the importance of the inflorescence phase in winter oilseed rape for the yield. However, the latest that the N dressing can be conducted is at the rosette stage [35]. 

Nitrogen fertilization in cereals, to meet the requirements at CK2, should be conducted in the period between the date of the growth rate change (transition point) and flowering (Figure 5). In fact, in cereals, the last dose of N is applied at the end of the stem elongation phase. This phase precedes the period of the highest rate of ear growth, i.e., booting, which is responsible for the number of grains per unit area [36,37]. A separate case is bread wheat, where the last dose of N is used during the heading stage. The main goal is to increase the protein content in the grain [38].

A relevant and crucial component of nutritional crop status evaluation is a well-defined range of nutrient concentration in indicative plant parts and the relationships between them. Theoretically, there are some sophisticated methods for crop nutritional status assessment. The most commonly used are DRIS (Diagnosis and Recommendation Integrated System) and CND (Compositional Nutrient Diagnosis) [39,40]. In practice, farmers use, the sufficient ranges (SR) method to gain a quick evaluation of the crop nutritional status [41]. The biggest disadvantage of the SR method is the need for a large data set that is required for the calibration of the established ranges [42]. Moreover, most of the current ranges used by farmers were generated in the past for crops yielding at much lower levels than today. Table 1 compares the SRs for maize and sugar beet at CK1. The presented ranges, in spite of elaboration in different regions of the world (Europe, USA), differ only slightly. This suggests their suitability for world-wide application. It is much more difficult to make a reliable assessment of the nutritional status of sugar beets or potato (Table 1). For example, the Bergmann’ sufficiency ranges developed at BBCH 41 for sugar beet are not currently suitable for correcting the nutritional status of currently grown varieties. The last date of this crop fertilization with N must precede BBCH 33 [43,44]. 

Maize has been subjected to in-depth studies on its nutritional status at the onset of flowering (Table 2). The presented ranges, despite different origin in terms of geographical region and publication year, differ only slightly. The biggest differences concern the content of Ca and K. The main reason for these variations is the calibration of plant tests under conditions of significant differences in the content of soil Ca and K in the area of the conducted research.

### 2.2. The Root System Architecture—RSA

This section may be divided by subheadings. It should provide a concise and precise description of the experimental results, their interpretation, as well as the experimental conclusions that can be drawn. The uptake of nutrients related to the incorporation of ions or molecules into the plant’s organism consists of a series of sequential processes that can be divided into three main ones: Movement of nutrients along the soil/plant continuum:
transport of ions/molecules from the soil solution towards the root surface,ingrowth of the root into soil patches rich in available nutrients;Transport of nutrients adsorbed on the root surface through the plasma membrane into the cytoplasm;Direct utilization of the nutrient in the root or its transport via the xylem to active plant tissues.

The processes mentioned in points 2 and 3 are extensively described in scientific books and extended reviews [16,47]. Here, we discuss the key processes related to root system growth during the growing season. The functions of the root system of crop plants can be considered from several points of view [48,49,50]:Anchorage of the plant in the soil;Water extraction from the soil to:
stabilize the shoot temperaturetransport nutrients to the shoot;Nutrient uptake from the soil solution;Impact on rhizosphere processes through:
release of organic compounds → a source of energy for microorganisms present in the rhizosphererelease of protons or chelating agents → increase in nutrient availabilitydeposition of carbon by dead roots → humus build-up;
Symbiotic associations with bacteria or fungi;Storage organs, treated as main yield (sugar beets, cassava, sweet potato).

The root system, despite seasonal dynamics and spatial variability, is a conservative trait of the plant. It can be characterized as a three-dimensional structure, creating the root system architecture (RSA) [51,52]. The components that describe RSA include three main characteristics of the root system:Primary root (PR) length, which determines the depth of a plant rooting;Root branching patterns, which are represented by a number of characteristics, among others (i) number of lateral roots (LR), number of adventitious roots (AR), (ii) growth angle of LR and AR in relation to the primary root (s), (iii) root diameter, (iv) root length density (RLD);Root hairs (RH), including length, diameter, number per root unit length or area.

Generally, on the basis of the plant branching patterns, the root systems of crops, that are botanically justified, are classified as taproots (dicotyledonous species, dicots) and fibrous roots (monocotyledonous species, monocots). The main components of the taproot system are PR, LS, and AR roots. The fibrous root system consists of PR, seminal roots, crown roots and AR roots [53].

The spatial distribution of roots in the soil profile is important for both the current rate of crop growth, as a decisive factor for the uptake of water and nutrients, and for maintaining soil fertility due to allocation of carbon. The spatial arrangement of the root system in the soil profile, in spite of its heterogeneity, can be described by specific parameters or indices. This concerns, first of all, the general shape of the root system profile down to the soil. The key parameters are: (i) distribution of the total root biomass, (ii) plant rooting depth, (iii) root length density [48,54]. Root distribution with depth can be best described using, for example, an exponential model by Gerwitz and Page [55]:(10)$$Y=A\left(1-{\;e}^{-\mathrm{cx}}\right)$$
where: Y—the cumulative fraction of roots between a soil layer of 0–10 cm and the depth x + 10 cm (cm); x—a defined soil layer below 0–10 cm; c—an empirical fitting parameter that determines the root distribution with depth. 

This equation or others, more mathematically advanced, are used to define the effective rooting depth (ERD) as the key RSA parameter [56,57]. Most of the root biomass is present in the topsoil, decreasing exponentially with the soil depth. As estimated by Fan et al. [56] for main crops grown in a humid climate 50% of the root biomass is in the top 20 cm. The remainder part of roots, present in the subsoil, is important for water and nutrient uptake. Under conditions of drought, the uptake of water and nutrients from deeper soil layers is critical for both growth and yield maintenance [58]. The ERD is defined as the potential depth of the soil profile from which plant roots can extract the maximum amount of water available to plants from the soil during dry years. The soil layer, extending between the soil surface and the ERD, is known as the effective root zone (ERZ) [59]. This zone, depending on the assumption, covers 80% or even to 95% of the total root mass or root extent. As reported by Fan et al. [56], 50% of wheat root biomass is present within 16.8 cm of the soil surface layer, while 95% reaches down to 103.8 cm of the soil profile. In agricultural practice, the ERD is used to assess both the water resources for the currently grown crop and/or the dose of irrigation water. For example, the ERZ in the Czech Republic is estimated at 80–100 cm for winter cereals, and at 40–50 cm for potatoes [60]. 

The role of the subsoil in plant growth and yielding is usually ignored in the diagnosis of crop plant fertilization. This ERD is, in fact, used as a routine diagnostic tool to determine the content of mineral nitrogen (N_min_). For most crops, this analysis is performed down to a depth of 90 cm [61]. Subsoil is an important storage of other nutrients, including P [62]. Current studies document that these P resources are used by plants, provided that the P balance in the topsoil is negative. This conclusion is probably the result of using powerful extractants to determine the available P [63]. A study by Barłóg et al. [64] clearly showed that extraction solution for N_min_ determination can also be used to determine the resources of other nutrients. As shown in Figure 9, the seasonal pattern of available P (0.01 M CaCl_2_ extract; soil: solution ratio as 1:5), regardless of the season (crop), was stable. The P content was in a declining pattern in the soil profile. With the exception of 2005, its content was lower at crop maturity compared to spring. These P resources can be exploited by plants to up to 60% of its total content in the 0.9 m soil layer [65]. 

### 2.3. Root System Growth during the Growing Season

The genetically determined root system of plants is heterogeneous both in time and in soil space [48,66]. The first variable is inextricably linked with the plant’s life cycle. Generally speaking, the growth of the root system as an integral part of the shoot system, is a result of both organs functional interdependences [67,68]. Maintaining a stable but temporary balance between the supply of water and nutrients to the shoot by the roots and the return supply of assimilates to the roots form the shoots is the basis that determines the plant growth rate, development of yield components, and yield [69]. 

The relationship between these two organs of the plant during its life cycle is not constant, as expressed by the ratio of the biomass of the shoot to the biomass of the root (S/R). Its value, as a rule, increases with plant growth (Figure 10). A frequently asked question concerns the relationship between shoot growth and the ability of the root system to supply the required amount of nitrogen [67]. In cereals, the highest rate of N uptake by roots occurs in the period from the end of tillering to the stage of full stem elongation (BBCH 29 to BBCH 37; Figure 6). For example, during this period, the rate of N uptake by winter rye plants on a plot fertilized with NPK and manure (long-term static experiment, existing 30 years before the study) was 3- and 10-fold faster compared to plants grown on a plot fertilized only with manure or on the absolute control [70]. Moreover, the rye root system on the NPK + manure plot was both shallower and more branched than on the absolute control [71]. This is in line with current studies on wheat, highlighting the importance of the early stages of stem elongation for the development of yield components [37]. Moreover, the period of the highest N uptake by winter rye confirms the well-defined CK1 (Figure 5).

The second variable affecting the RSA concerns the impact of soil and environmental conditions on the development of the root system during the growing season. The primary factor of root growth is temperature, which determines the rate of all metabolic and physiological processes during a plant’s life cycle [12]. The optimum temperature for root growth is much lower for plants from temperate than tropical climates [69]. The second factor is water, the function of which, similarly to temperature, cannot be separated into individual processes [72]. The third factor is soil fertility, which determines the efficiency of water and nitrogen [5]. The effect of soil fertility on RSA depends on the course of temperature and water conditions during the growing season. Any change in the environmental conditions for the worse (temporary water shortage, lower level of soil fertility, low availability of nitrate nitrogen) increases the plant’s input into the root system size, mainly increasing its rooting depth—the primary root and root hair length, while reducing the development of lateral roots. The observed morphological changes are due to the actions of hormones, which are dependent on the availability of nitrate nitrogen [73,74]. 

The maximum demand for nutrients by a plant, as shown in Figure 4 and Figure 5, occurs during the linear phase of the biomass accumulation by the crop. Soil inherent (quasi natural) resources of nutrients can be potentially high, but the plant’s nutrient requirements at the maximum growth are higher than their supply to the plant from soil solution [8]. The rate of any given nutrient movement in the soil solution towards the root surface depends, among others, on the value of its diffusion coefficient. In pure water, the differences between diffusion coefficients for nutrients are small compared to their values in the soil solution (Table 3). The coefficients for NH_4_^+^ and K^+^ are about 100-fold lower compared to the nitrate ion (NO_3_^−^). An even lower value is the attribute of the orthophosphate ion. Moreover, the differences between the values of the diffusion coefficients for all these ions increase with the decrease in the water content in the soil [75]. 

The absorption of a given nutrient by the plant root results in a decrease in its concentration around the root. This phenomenon is called the depletion zone, which is specific for each individual nutrient [67]. The size of the nutrient depletion zone (NDZ) is determined by two key variables: (i) value of its diffusion coefficient (D_eff_); (ii) soil exploitation time by the root (t). The influence of both variables on NDZ can presented in the formula:(11)$$\mathrm{NDZ}={\left(2\times {\;D}_{\mathrm{eff}}\times \;t\right)}^{1/2}$$
where: NDZ—the size of the depletion zone, cm; D_eff_—diffusion coefficient of a particular nutrient, cm^2^ s^−1^; t—time, s. 

The NDZ arises when: (a)metabolic requirements of the above-ground parts of a plant are higher than the rate of nutrient supply to the plant from the soil solution;(b)the effective diffusion of a nutrient is sufficiently high;(c)the time of the plant root interaction with the soil in a particular soil zone is long enough.

The uptake of nutrients during the Yield Formation Period (YFP) of plant growth depends on the rooting depth and the root length density (RLD, cm cm^−3^) of the growing crop. The effect of RLD on the size of NDZ is nutrient specific. The rate of nitrate nitrogen ion (NO_3_-N) movement to the root is the most rapid of any nutrient, resulting in the fastest increase in NDZ around the root. Competition between neighboring roots occurs when their density exceeds 1–3 cm cm^−3^. Maize with an RLD of 3 cm cm^−3^ absorbs about 70% of NO_3_-N present in the soil solution. At the same time, the degree of P and K depletion does not exceed 5% and 10%, respectively [78]. Competition between the roots for P may occur, provided the RLD exceeds 30 cm cm^−3^ [79]. The RLD for crop plants rarely exceeds 2–5 cm cm^−3^. The exception are grasses, for which this parameter is in the range of 3–20 cm cm^−3^ [79]. An apparent paradox for both traits of RSA is that the greatest RLD values in the topsoil, regardless of the crop, decline exponentially with depth [67]. Current studies on winter wheat and winter oilseed rape have shown that the critical RLD of 1 cm cm^−3^ was 32 and 45 cm, respectively [80]. These data indicate that there is no competition between roots for NO_3_-N below this depth. However, it can be assumed that the presence of roots in the deeper soil layers has a significant impact on the yield. As recently presented by Grzebisz et al. [35], the content of NO_3_-N in the soil layer (0.6–0.9 m) was the key nutritional factor that determined the yield of winter oilseed rape (WOSR). As shown in Figure 11, the greater the decrease in the NO_3_-N content during YFP, the greater the WOSR yield obtained. During YFP, the sequential application of N_f_ creates rich N-NO_3_ zones in the topsoil, while the deeper soil layers are, as a rule, much poorer in nitrate content. Nevertheless, no reduction in root growth is observed within this mega-phase, either in the topsoil or the subsoil [71,81]. The ingrowth of the primary root in the subsoil and the simultaneous growth of lateral roots in the rich NO_3_-N niches in the topsoil can be explained by the *foraging strategy* of a crop [54,82]. This phenomenon entails the synchronization of both the local and systemic signals within a plant in response to the NO_3_-N status in the soil profile. The decrease in concentration of NO_3_-N in the subsoil, which is a typical phenomenon during YFP, leads to the increased flow of auxin to the apex of the primary root. As a consequence, it stops the growth of lateral roots within a soil zone poor in nitrates. At the same time, the induced systematic signal released by the apex of the primary root results in a compensatory growth of lateral roots in soil zones rich in nitrates [83]. The application of N_f_ by the farmer during the growing season leads to the formation of soil zones—*foraging patches* for a plant, which temporarily differ in the concentration of NO_3_-N. Therefore, it can be concluded that a split N fertilization system is a useful way to increase the efficiency of the applied N_f_. 

## 3. Soil Factors Affecting FUE

### 3.1. Soil Texture

The most important soil physical properties include: soil texture, density, structure, porosity, consistence, temperature, air and color. Among them, soil texture is the basic physical feature that determines not only the other physical properties of the soil, but also the chemical ones [84]. The percentage and mineralogical composition of the smallest mineral fractions in the parent rock determines the primary soil potential to supply plants with nutrients, which is the function of weathering and transforming primary minerals [85]. In addition, the content of mineral colloids is positively correlated with soil organic matter (SOM), which in turn is a source of organic colloids, which have a great impact on the water retention of the soil, cation exchange capacity, erosion processes, as well as soil microbial activity [86]. SOM sequestration is achieved through various mechanisms which include the formation of clay-humic complexes, sorption of organic matter on clay particles, fixation of organic carbon in the crystal lattices of clays and the formation of organo-metallic compounds such as Ca, Fe and Al humates through humification processes [87,88]. In general, the greater the SOM concentration, the greater the sorption capacity of the soil, and potential for water retention in soil [89] and nutrients [90]. Numerous studies show that soils with a high proportion of clay particles have a higher content of nutrients than soils with a low content of nutrients, not only in terms of general forms, but also plant-available forms [91,92]. At the same time, the clay content affects the fixation and de-fixation processes of some nutrients, especially K^+^ [91]. On the one hand, excessive fixation reduces the pool of mobile K^+^ ions in the soil and reduces the use of potassium from fertilizers, especially in dry soil conditions. On the other hand, it prevents the leaching of potassium from the soil [93]. Moreover, adsorption and non-exchangeable ammonium nitrogen (NH_4_^+^) fixation in soil is highly dependent on clay mineral composition [94]. Another problem with soil texture is water infiltration and the leaching of nitrates (NO_3_^−^) resulting from ammonium nitrification. Coarser-textured soils are more susceptible to soil N loss following the leaching of NO_3_^−^, and thus have potentially lower FUE values [64]. Furthermore, soil texture largely affects fertilizer and soil P transformations in soils. In coarser-textured soils the content of labile P fractions after adding phosphorus fertilizers is higher than in clay and loam soils. Therefore, in these soils there is a high risk of P transfer from soil to water systems [95].

### 3.2. Water Content

One of the most important factors controlling nutrient uptake and utilization by plants is the water content of the soil. First of all, water determines the processes of nutrient release from the soil solid phase to the solution phase [96,97]. Water deficiency in soil negatively affects microbiological activity and the processes of mineralization/biological fixation [98]. Water is also essential for dissolving and releasing nutrients from mineral fertilizers, including controlled release fertilizer [99]. However, from the point of view of the process of uptake of nutrients by plants, two phenomena deserve special mention: mass flow and diffusion [100]. Water deficiency in the soil reduces the intensity of both processes, and thus leads to a reduction in the amount of nutrients flowing to the root surfaces [101]. In this aspect, the degree of plant reaction to water stress depends on the element and its function. According to Oliveira et al. [102], in maize the proportion of mass flow contribution to Ca, Mg, N, S and K transport was as follows: 100, 63, 56, 45 and 10%, respectively. This series clearly shows that the supply of plants with Ca and Mg may be severely limited in drought conditions, despite their relatively high concentration in the soil compared to other macronutrients [103]. Taking into account the diffusion processes, a water shortage in the soil will primarily limit the mobility of phosphate ions and micronutrients. Moreover, it will lead to the intensification of precipitation processes and the crystallization of amorphous compounds of phosphorus with other cations, depending on the pH [104]. As the water deficit in the soil increases, the proportion of pores filled with air increases, mechanical resistance increases, and the rate of root growth decreases. Under conditions of high soil oxygenation, the potential of the soil to supply plants with some micronutrients is reduced (Fe, Mn), whose higher oxidation state forms are less plant-available than the reduced forms [105]. The second group factors effecting NUE directly relates to the plant response (growth) and its ability to convert in biomass the assimilated/remobilized nutrients, especially nitrogen [106]. Water has a direct effect on root growth. In order to meet the demand for water, the roots constantly explore the soil, building a very complex, branched architecture [107]. An increase in the number of hairs and diameter root tips has been observed in plants under drought conditions. Root hairs greatly increase root-soil contact and the surface area available for adsorbing water and nutrients [108]. However, dense and deep root systems are not always good under all hydrological conditions, for example they poorly capture water from the topsoil under low rainfall conditions [109]. In drought conditions, the above-ground mass is reduced more than the underground mass, which in the case of a long-lasting drought may limit the inflow of assimilations and stop root growth, with all the negative effects of this phenomenon [110]. Lupini et al. [111] reported that water stress in durum wheat reduces the values of NUE, NUpE, and NUtE indices, regardless of the genotype. However, it should be remembered that excess water is just as harmful to plants as is its deficiency. One of the reasons for this is the reduction in the oxygen content in the soil needed for the respiration of plants and microorganisms [98]. In addition, large amounts of iron or manganese are released, which in excess may be toxic or interfere with the absorption of other nutrients. This phenomenon is particularly harmful in the cultivation of rice paddy on acidic soils [112].

### 3.3. Soil Compaction

Another important physical factor influencing nutrient uptake from soil, as well as their utilization from fertilizers, is soil compaction. Compaction affects plant growth by reducing the content of soil air and plant-available water, and the consequent restricted root growth results in the plant being unable to obtain an adequate amount of nutrients. Soil compaction can be assessed by measuring the following soil properties: bulk density, porosity and mechanical impedance [113]. Mechanical impedance is defined as a physical barrier to developing roots as a result of excessive bulk density. In general, root growth rates decrease sharply for soil mechanical impedance values between 0.8 and 3 MPa. On the other hand, when assessing soil compaction by soil bulk density, most authors give the value of 1.47–1.85 g cm^−^^3^ as critical for crops, depending on the percentage of clay [114,115]. The turgor in the cells in the elongation part of the roots determines their ability to overcome the mechanical resistance of the soil [116]. The greater it is, the greater the probability of root growth into the zone of compacted soil [117]. At the same time, root elongation is facilitated by root secretions and abraded side cells of the roots, which reduce the effect of the friction force [118]. When the mechanical resistance is too high, changes are observed at the physiological level (accumulation of solutes, reduction in the growth rate, new cell production) as well as anatomical (increase in the root diameter and the share of mechanical tissue in the direction of growth) [119,120]. The entire root system develops into less resistant parts of the soil, often forming a shallow system with the roots parallel to the soil surface [121]. According to Ramalingam et al. [122] the root length density at 30–60 cm soil depth decreased with hard compaction (to 70% of control) and increased with moderate compaction (to 135%). At the same time, the number of roots with a deep angle (i.e., 45° to 90° from the horizontal) correlated with the root length density and its proportion was lower in compacted soil. Considering the root architecture, the studies carried out so far have shown that deeper root growth is more important for N uptake than increased root density [123]. In this respect, it is necessary to remove the soil compaction in the subsoil. On arable land, the use of heavy machinery increases the risk of soil compaction especially in the subsoil [124]. Changes in the root architecture mean that the plant is unable to fully use nutrients, especially those whose main reservoirs are in deeper layers of soil [125]. Regardless of the soil depth, when the soil is characterized by excessive bulk density and/or mechanical impedance, the roots develop mainly in macro-pores [126]. This results in a poor supply of nutrients in plants under soil drought conditions, as the macro-pores in soil water retention only contribute to a small extent [98]. Another important issue with soil compaction is the loss of nitrogen from the soil through the emission of its gaseous forms into the atmosphere. As a result of soil compaction and the oxygen deficiency caused by this process, the activity of denitrifying bacteria increases and the production of N_2_O and N_2_ increases [127]. The emission of these gases to the atmosphere is favored by the low values of the parameters that define gas diffusivity in compacted soils [128]. According to Ruser et al. [129], high N_2_O emissions in compacted soils occurred at a water-filled pore space > 70%. N_2_ production took place only at the highest soil moisture level (>90% water-filled pore space) but it was considerably less than the N_2_O-N emission in the most compacted areas in a potato field. Soil compaction also increases the volatilization of ammonia, as compared to uncompacted soils [130]. However, for this gas, the emissions are mainly determined by other soil physical and chemical characteristics [131]. 

### 3.4. Soil Temperature

Temperature has a substantial effect on some soil properties as well as root growth. Important processes depend on the temperature of the soil, such as: soil structure, aggregate stability, soil moisture content and aeration, soil pH, cation exchange capacity (CEC), soil microbial activities and organic matter decomposition [132]. A soil temperature between 2–38 °C increases the decomposition of organic matter by stimulating microbial activities and increasing the solubility of chemical compounds [133]. As a result of decomposition, the resources of N, P, S and other nutrients available to plants increase [134]. From the point of view of the nutritional status of plants, an extremely important temperature-dependent process is the availability of P to plants. Soils with low temperature have low availability of P because the release of P from organic material is limited [135]. Soil temperature also influences the P diffusion coefficient in the soil. Yilvainio and Pettovuori [136] observed that water-soluble P increased with soil temperature from 50 to 250 °C due to the increase in the movement of P in soil controlled by diffusion. Soil temperature also affects nutrient uptake by changing soil water viscosity and root nutrient transport. At low soil temperature, nutrient uptake by plants is reduced as a result of high soil water viscosity and low activity of root nutrient transport [137]. In general, low temperature decreases both root elongation and branching. However, low temperatures inhibit shoot growth more than root, leading to a high root/shoot dry matter ratio [138]. Vessel lignification can be delayed and axial hydraulic conductivity is higher in roots grown at low temperatures compared to high temperatures [139]. Thus, tomato, for example, showed that low soil temperature results in reduced root growth, tissue nutrient concentrations and, as a consequence, the amount of the component taken from the soil [140]. The unfavorable effect of higher temperature is marked in various ways. Too high a temperature may lower the CEC, and at the same time cause an increase in the concentration of hydrogen protons (increase in soil acidification) due to the high rate of soil organic matter decomposition. The plant’s response to temperature changes depends not only on the plant species, but also on the content of nutrients in the soil. According to Xia et al. [141], negative effects of excessive temperature on P content and uptake occur especially in P-poor soils. The authors also found that an overly high root zone temperature reduced root vitality and plant phosphorus content, which in turn affected plant growth and light energy utilization efficiency. 

### 3.5. Soil Reaction 

Among a number of chemical parameters describing the chemical properties of soils, the use of nutrients from fertilizers is very much influenced by its pH [142]. This feature directly relates to the concentration of active H^+^ protons in aqueous solutions, and indirectly it is a measure of the acidity or alkalinity of a soil. The influence of soil pH on the nutrient uptake of plants results from many different phenomena and processes. The most important ones include: effecting the content of plant-available forms of nutrients in soil; capacity and proportions between cations in CEC; activity of trace elements and heavy metals; soil microbial activity, biological N_2_ fixation; emissions of ammonia and other gases from the soil [143,144]. Both too acidic and alkaline soils have a negative effect on nutrient uptake. However, the phenomena occurring in acidic and alkaline soils differ significantly in terms of processes contributing to their degradation. A significant problem of acidified soils is an increase in exchangeable aluminum (Al^3+^) [145]. The content of this form of aluminum monomers rapidly increases in soils below pH 5.0–5.5 ([146]; Figure 12). An excessive amount of Al^3+^ ions in the soil negatively affects the nutrient uptake processes and plant growth [147]. Numerous studies show that even at the stage of nutrient uptake, unfavorable phenomena take place, such as the competition of Al^3+^ ions with other ions for attachment sites in the apoplast, in carriers, attachment to the ATPase of cytoplasmic membranes and disruptions in the operation of the proton pump [148,149]. An excessive content of Al^3+^ ions in the soil significantly reduces the uptake of Mg^2+^ ions. This is due to the similar size of the hydrated ions [150]. One of the most important consequences of the presence of exchangeable aluminum in the soil is the disturbance of the growth and development of the root cap, and consequently the shortening of the root length and unfavorable changes in its structure [151]. For most crops, even a small concentration of exchangeable aluminum (in nanomoles) in the root cells is a toxic factor for the metabolic, physiological, genetic and biochemical processes taking place in the plant [152]. The reduction in the root system negatively affects the use of nitrogen in fertilizers and increases the risk of nitrate being washed out from the soil [153]. Moreover, nitrate nitrogen, which is not taken up by plants, is reduced to gaseous compounds, including N_2_O [154]. In highly acidic soils, apart from exchangeable aluminum, excessive amounts of manganese (Mn^2+^) and iron (Fe^2+^) can also appear, which can further disrupt the proper growth and development of plants [155]. 

### 3.6. Soil Salinity

In arid or semi-arid climates, the problem is not soil acidification, but alkalization and salinity [156]. Under low rainfall conditions and a high evaporation rate, Na^+^ ions, as well as various soluble salts, accumulate in the soil. Their excessive accumulation contributes to the significant advantage of OH^−^ ions over H^+^ and, consequently, to an increase in soil pH to the level of 9–10 [157]. Soil alkalinity can also be increased by the addition of water containing dissolved bicarbonates, especially when irrigating with high-bicarbonate water [158]. The low osmotic potential of water in saline soils adversely affects water absorption by plants and nutrient uptake [159]. Salinity of soil significantly decreases P uptake by plants because phosphate ions precipitate with Ca ions contained in saline soils [160]. However, the alkalinity of soils is most often associated with the Na concentration [161]. Alkaline soils are characterized by unfavorable physical conditions, low content of plant-available forms of microelements and phosphorus, components determining nitrogen metabolism in the plant. During nutrient uptake processes, Na^+^ ions compete for carriers with other nutrients in cationic form, in particular with K^+^ ions [162]. This is a negative phenomenon because Na, dissimilar to K, negatively affects the activity of plant enzymes [163]. The reduced uptake of K^+^ ions also means the insufficient or slower transport of NO_3_^−^ from the roots to the above-ground parts, and thus poor efficiency of N from fertilizers [164]. Furthermore, an excess of Cl^−^ ions in the soil has a negative effect on NO_3_^−^ uptake. However, as recent studies show, optimal NO_3_^−^ vs. Cl^−^ ratios become a useful tool to increase crop yield and quality, agricultural sustainability and reduce the negative ecological impact of NO_3_^−^ on the environment and on human health [165]. Under saline soil conditions, plants change their root architecture, which also has negative consequences for nutrient uptake [166].

### 3.7. Soil Organic Matter

The content of soil organic matter (SOC) in soil is one of the most important features influencing soil fertility [167]. Changes in SOC are associated mainly with changes in macronutrient contents, such as N, P and sulfur (S) which are chemically bound to carbon (C) in organic compounds [168]. Therefore, in systems where SOC content is declining, soil fertility declines over time and soils become increasingly dependent on the use of mineral fertilizers, especially nitrogen [169]. A total loss of organic N directly translates into a weaker potential of soils to release mineral forms that are taken up by plants. At the same time, under such conditions, the demand for N from fertilizers increases. Numerous experiences show that the most effective use of N from fertilizers is observed in the small dose range [9,170]. Conventional tillage with plowing can reduce SOC stocks by 30–60% [168]. Changes in NUE resulting indirectly from the increase in the degree of SOC degradation are confirmed by research of Luis et al. [171]. The authors calculated that over many years the efficiency of nitrogen fertilization application decreased from 68% in 1961 to 47% in 2010. This means that the use of N from fertilizers deteriorated and N losses to the environment increased by 21%. 

In general, the transformation of native soil to agricultural uses leads to a decline in SOC levels [172]. However, agricultural land uses do not always result in losses of SOC. The rate and direction of changes in the C content in soils depend on the soil use system, irrigation, crops, and the level of organic matter return to the soil [173,174]. Failure to plow or use various simplified systems leads to the accumulation of SOC, especially in topsoil [175]. Reduced tillage in comparison with ploughing increased SOC stocks in the surface layer (0–10/15 cm) by 20.8% or 3.8 t ha^−1^, depleted SOC stocks in the intermediate soil layers to 50 cm soil depth with a maximum depletion of 6.6% or 1.6 t ha^−1^ in 15/20–30 cm and increased SOC stocks in the deepest (70–100 cm) soil layer by 14.4% or 2.5 t ha^−1^ [176]. However, the use of natural and organic fertilizers is of greater practical importance in maintaining an appropriate SOC [177]. Szajdak et al. [178] reported that a yearly application of 30 t ha^−1^ of manure to light soil over 38 years doubled the SOC content. The increase in plant biomass as a result of the use of NPK fertilizers leads to an increase in the influx of C to the soil. Nevertheless, accumulation of C in soil is not favored by an excess of N in the soil from high fertilizer application rates and/or low plant uptake can cause an increase in the mineralization of organic carbon which, in turn, leads to an increased loss of C from soils [179].

### 3.8. Nutrient Shortage

Factors responsible for nutrient deficiency in crops can be divided into two main groups: (i) causing an absolute deficiency of nutrients in soil, resulting from low nutrient contents in the parent soil material, low level of SOC, nutrient losses from the soil, e.g., Mg leaching, long-term unbalanced crop fertilization practice neglecting nutrient depletion in soils through crop nutrient removal; (ii) causing an induced deficiency, resulting from factors that disturb the flow of nutrients to the root such as: improper moisture and temperature of soil, ion competition, factors responsible for root system size, etc. [12]. The natural source of most nutrients in the soil are primary and secondary minerals. As a result of weathering, often stimulated by the activity of living organisms, potentially available nutrients are released into the environment. Their reactions in soil and fate in the environment depend on the type of element. Some nutrients are strongly absorbed in the soil, others are easily lost (by leaching or emission). The first group includes K. The ions of this element can be absorbed in the soil in an exchangeable and non-exchangeable form [180]. The second type of adsorption prevents the elution of K^+^ ions from soil, but on the other hand this leads to a reduction in the potential to supply plants with K. This phenomenon is responsible for the poor efficiency of K from fertilizers on soils rich in mineral colloids. The strength of the non-exchangeable K ion fixation increases in dry years, which further aggravates the symptoms of water stress [181]. Non-exchangeable adsorption may also apply to other cations, e.g., Mg. However, in relation to Mg, the degree of soil moisture has a greater practical importance, as this element is assimilated by plants as a result of a mechanism known as mass flow. Contrary to K, Mg is less readily absorbed [103]. This is one of the reasons for the relatively easy leaching of Mg from the soil. An absolute deficiency of K and Mg leads to a poor efficiency of N, as both elements greatly affect the metabolism and transport of N in plants [16]. Studies conducted on sugar beet show that Mg applied to the soil significantly increases the agronomic efficiency of N, but in the range of low doses of N (Figure 13). This indicates that an excess of nutrients in the soil may not lead to better FUE/NUE values. Phosphorus and micronutrient deficiencies in the soil are often the result of an inappropriate pH range in the soil. In an acidic reaction, the adsorption of P on iron and aluminum compounds increases, while in an alkaline pH, insoluble calcium phosphates precipitate in the soil [104]. An inappropriate soil pH also influences, directly or indirectly, the content of plant-available forms of K, Mg and Ca [144]. Thus, in order to restore the optimal conditions for the uptake of nutrients, it is necessary to regulate and/or constantly control the soil pH. If this does not help, then one option is to enrich the soil with nutrients to eliminate their absolute deficiency, or to support the plants by foliar fertilization.

## 4. Innovations on the Fertilizer Market

Innovations in the fertilizer market involve two main areas of research activity. The first one concerns the process of obtaining raw materials and the production of fertilizers. The production of fertilizers, especially nitrogen ones, is energy-intensive and is a significant source of greenhouse gases. In this context, two strategies for the production of ammonia are considered: blue hydrogen—steam methane reforming with carbon capture and storage (CCS) and green hydrogen—electrolysis of water, to generate hydrogen and oxygen in a process driven by sustainable energy [183]. The second area of fertilizer production, mainly aimed at improving NUE indicators, concerns a number of application aspects and the chemical composition of fertilizers. Research shows that about 40–70% of N, 80–90% P and 50–70% of K from fertilizers is lost to the environment and cannot be used by plants, thus posing a threat to the environment [184]. For many years, the fertilizer industry has been improving and introducing Slow-Release Fertilizers (SRF) and Controlled-Release Fertilizers (CRF) [185]. 

The advantages of nitrogen fertilizers from the SRF and CRF groups derive from the following features: (i) they ensure a good supply of nitrogen to plants, especially in critical phases; (ii) they reduce the number of application rates; (iii) they reduce the nitrate content in plants; (iv) they limit nitrogen losses and reduce its negative impact on the environment [186,187]. With regards to nitrogen fertilizers from the SRF group, the delay of action is achieved by the formation of slightly soluble compounds, most often polymers based on urea and aldehydes, e.g., formaldehyde [188]. The condensation products of urea and other compounds can be used as solid (e.g., ureaform) or liquid fertilizers (e.g., urea-triazone). Research shows that liquid slow-release nitrogen fertilizer increases yields and nitrogen use efficiencies (NUE) in rape plants compared with a standard urea fertilizer [189]. For the production of CRF fertilizers, highly water-soluble compounds are used. Dissimilar to SRFs, controlled-release fertilizers (CRFs) are less influenced by soil temperature or texture, and they are not so dependent on soil microbiology [190]. The effect of delaying N release is achieved by covering the granules with a different type of protective layer (e.g., sulfur coatings, polystyrene, polyethylene, polyurethane, polysulfone resin and waxes coatings, siloxanes, etc.) [191,192,193,194,195]. The protective layers prevent the inflow of water from the soil to the inside of the granules and the dissolution of the contained compounds. The positive effects of different coated urea fertilizers on crop yield and NUE have been observed by many authors [196,197,198]. Recently, a great deal of attention has been paid to fertilizers using biochar and lignite for coatings, as they allow for the cheap production of CRF fertilizers [186]. Additionally, carbon-based materials, which contain humic acids act on plants such as biostimulants. The results of Wen et al. [199] also suggest that biochar-based slow-release nitrogen fertilizers could significantly improve the water-holding and water-retention capacity of soil. As a result, on the field scale, in rice cultivation, the optimal dose of N in the form of CRF (coated with lignosulfonates) fertilizer was 20% compared to using traditional nitrogen fertilizer [200]. In turn, according to Ghafoor et al. [198], biochar-based CRF fertilizers effectively reduce the nitrogen-release rate (69.8% of nitrogen was released after 30 days) and possess low nitrogen-leaching-loss amounts (10.3%), low nitrogen migrate-to-surface-loss amounts (7.4%), and high nitrogen-use efficiency (64.27%), as compared to other N fertilizers, consequently effectively promoting cotton plant growth. According to Guo et al. [201] fertilization of maize with CRF fertilizer with the addition of humic acids allows not only an increase in NUE, but also significantly reduces the emission of N_2_O from the soil to the atmosphere by 29.1–32.6% compared to CRF fertilizer without humic acids. For the production of CRF fertilizers, nitrogen stabilizers are also used: urease and nitrification [185]. Among the various urease inhibitors, the most commonly used are N-(n-Butyl) thiophosphoric triamide (NBPT) and N-(n-propyl) thiophosphoric triamide (NPPT). The most commonly applied nitrification inhibitors are: 2-chloro-6-(trichloromethyl) pyridine (nitrapyrin), dicyandiamide (DCD) and 3,4-dimethylpyrazole phosphate (DMPP). The literature shows that the application of these inhibitors has considerably reduced inorganic N leaching, N_2_, NO and N_2_O emission while at the same time improving crop yield and N use efficiency [202,203] However, their effect on yield is very variable and depends on many factors. The application of fertilizer with urease inhibitors can increase the content of ammonium nitrogen in the soil by 10–59% compared to treatments without these inhibitors [204]. According to some researchers, it may increase the N resources in the soil and, consequently, gas losses of N (NH_3_, N_2_O) from the soil [205]. Therefore, the combined application of urease inhibitors with nitrification inhibitors reduces multiple losses associated with volatilization and denitrification [206]. Urea with urease and nitrification inhibitors can be used simultaneously to improve the N uptake, seed yield and grain protein contents, for example in quinoa [207]. Meta-analysis by Yang et al. [208]. showed that among the popular nitrification inhibitors, DCD was more effective than DMPP on increasing plant productivity. An increase in crop yield by DMPP was generally only observed in alkaline soil. This is confirmed by the results of Alonso-Ayuso et al. [209], who on soil with a pH of around 8.0 obtained after DMPP application allowed a 23% reduction in the fertilizer rate without decreasing maize yield and grain quality. 

With respect to physical characteristics, in recent years urea has been produced with larger granules, facilitating mixing with fertilizers of similar grain size and bulk density, and allowing a wider spreading width compared to traditionally granulated urea. This is especially suitable for the fertilization of rice [210].

Innovations in the phosphorus fertilizer market also include the production of fertilizers with a controlled phosphorus release rate (CRFs). Their use increases the efficiency of using P from fertilizers (PUE) compared to traditional phosphorus fertilizers, and at the same time they reduce the negative impact of fertilization on the environment [191,211]. The rate of phosphorus release from fertilizers depends on a number of factors, including type and thickness of coating material, soil temperature and pH, humidity and microbial activity [193,212]. According to Fertahi et al. [213], 3 days after the application of phosphorus fertilizers, 100% P was released from water-soluble triple superphosphate (TSP) granular fertilizers, and only 60% from biopolymer coated TSPi. Next, Barbosa et al. [214] reported that biochar-based phosphate fertilizers have potential as a support material to increase the availability and efficiency of N use by plants. It should be mentioned that phosphorus fertilizers with a controlled phosphorus release rate also include: partially acidulated phosphoric (PAPR) and thermophosphates [215,216]. Stabilization of phosphorus transformations in the soil, and thus an increase in the potential P uptake from fertilizers, can be achieved by adding chemicals, the so-called phosphate boosters [217]. Their task is to decrease P-adsorption in soil and increase soluble-P from applied fertilizer-P [218]. Another solution for the future increase in PUE may be the addition of solubilizing bacteria [219]. In the foliar fertilizer market, fertilizers containing P in the form of phosphonates are now available. They have a beneficial effect not only on the nutritional status of plants, but also on their tolerance and resistance to fungal parasites [220]. 

The introduction of amino acids or other organic compounds of a biostimulating nature to the composition of fertilizers has also been a breakthrough in the foliar nutrition of plants. Amino acid molecules, distinct from technical salts or synthetic chelates, are electrically neutral, therefore the assimilation time of nutrients from fertilizers is short and their use will improve the nutrient use efficiency compared to traditional foliar fertilizers [221,222,223]. Glutamic acid has a particularly strong complexing effect [224]. On the other hand, some ammonium acids show a typical biostimulating character; for example, tryptophan, which is an auxin precursor. As demonstrated by Gondek et al. [225], NPKS soil fertilizer with the addition of thryptophan increased the maize biomass and the use of N and S from the fertilizer by 27% and 17%, respectively, compared to fertilizer without an amino acid. The incorporation of various organic and mineral substances into the soil together with fertilizers is an important way to improve the efficiency of using nutrients from fertilizers. As reported by Palanivell et al. [218], clinoptilolite zeolite application could contribute to an improved use of nitrogen, phosphorus, and potassium fertilizers to prevent soil, air, and water pollution. This treatment also improved nitrogen, phosphorus, and potassium use efficiency. The use of slow-release fertilizer hydrogels (SRFH) is also of interest. SRFHs are a combination of a super absorbent hydrogel (SAH) and a fertilizer with both water retention and slow-release properties [226]. Polymer super-absorbents are macromolecular compounds capable of absorbing water or physiological fluids in amounts much greater than their mass. They can be added to the soil or to fertilizers [227,228]. Among other things, chitosan-based hydrogels can be used as an additive to fertilizers [229].

The application of nanotechnology to the development of new types of fertilizers is considered to be one of the most promising options to significantly increase global plant production without negatively affecting the environment [230]. According to the European Commission [231], “Nanomaterial” means a natural, randomly generated or manufactured material containing particles in a free state or in the form of an aggregate or agglomerate in which at least 50% or more of the particles in the numerical particle size distribution have one or more dimensions in the range of 1 nm–100 nm. Nanoparticles are 100 to 1000 times larger than the size of the individual ions of nutrients that are involved in biochemical reactions [232]. However, they are in dimensions similar to or smaller than a number of anaotomous structures of plant tissues, e.g., plasmodesmata, cell wall pore sizes, or stomates [233]. Therefore, the presence of nanoparticles in foliar fertilizers improves the bioavailability of nutrients due to the nano-size, large specific surface area and greater reactivity of the compounds [234]. Fertilizers applied to the soil create the possibility that particles in the “nano” size may not be easily fixed between sheets of secondary minerals, and so not easily leached away from the soil [235]. The advantages of nanofertilizers also include the application of nutrients in a relatively smaller amount, ultimately reducing the cost of transport and at the same time improving the ease of application [236]. Nanofertilizers are usually divided into three groups: (i) classic fertilizer, but containing nano-scale particles; (ii) classic, traditional fertilizers with the addition of fertilizers in the form of nanoparticles; (iii) nanoscale coating fertilizer, referring to nutrients encapsulated by nanofilms or intercalated into nanoscale pores of a host material [237]. The nanocarriers used in the last group, such as zeolites, chitosan, clay and other nanomaterials, can provide plants with an even release of macronutrients during vegetation, which in the case of nitrogen and phosphorus improves their use in fertilizers [238,239]. Currently, the market of foliar fertilizers is developing intensively, which, apart from traditional compounds containing microelements, also contain noble metals, in particular silver ions, showing specific properties as pesticides on the “nano” scale [233,240]. Despite the large amount of literature on the potential use of nanofertilizers, there is little credible scientific evidence to demonstrate their advantage over traditional fertilizers. According to Kottegoda et al. [241] application of urea-coated hydroxyapatite nanohybrids (HA–urea) results in the enhancement of nitrogen use efficiency and reduces the environmental impacts of rice cultivation. Raguraj et al. [242] reported an increase in tea yield by 10–17%, while reducing the urea dose by 50% compared to traditional urea. Li et al. [243] reported that application of P in the formulation of nanoscale hydroxyapatite (nHA) had beneficial effects on soybean P and Ca content upon high precipitation intensities. However, the authors did not record any significant difference in the effect of fertilizers on the soybean biomass. A meta-analysis by Kah et al. [244] found that the median efficacy gain of nanofertilizers over conventional fertilizers was 19, 18 and 29% for categories of macronutrients, micronutrients and nanomaterials acting as carriers for macronutrients, respectively. However, Kopittke et al. [245] are critical of these results. The authors note that numerous researchers describe the positive aspects of nanofertilizers, but the experiments often lack an appropriate control object that would allow an objective assessment of their effects on plants. In terms of the potential use of nanofertilizers in the future, carbon nanotubes (e.g., consisting of 60 atoms of C-fullarens), which may contain nutrients, mainly microelements, or other bioactive compounds, are of interest [230,235,246]. As a result of such a formulation, future nanofertilizers will fully meet the criteria of CRF fertilizers.

## 5. FUE—A Message for Agricultural Practice 

The stagnation in the increase in the crop yields is well-documented [247,248]. Despite considerable progress in breeding and the continual release of new varieties, the real improvement in NUE is small [249]. The challenge for the farmer to exploit the yield potential of the grown variety is:Determine the maximum attainable yield (Y_attmax_). This is the basis for choosing the most suitable variety for the actual climatic and soil conditions of the farm;Identify soil conditions that constrain:
growth and architecture of the root systemwater and nutrient availability;
Divide the whole field area into units of homogenous productivity;Identify *Nitrogen Hotspots* both on the farm and on the specific field;Observe the viability of plants at stages preceding the cardinal phases of yield formation;Schedule the correction of the plant nutritional status during the season to exploit its yield potential.

The effective control of the set of factors indicated above is crucial to optimizing NUE. The general formula can be written as: (12)$$\mathrm{NUE}=\frac{\mathrm{Nitrogen}\;\mathrm{Fertilizer}\;\mathrm{Rate}\;}{\mathrm{growth}\;\mathrm{factors}\;}$$

The denominator includes all growth factors that determine the plant’s uptake and utilization available N present in the soil-plant system during the growing season. The fractional value of all these factors, excluding N, should be ≤1.0 [5,8]. If the fractional value of a given growth factor approaches 1.0, its negative impact on NUE decreases and vice versa. The main challenge for the farmer in using N_f_ efficiently is to mitigate, or rather eliminate, the cause that leads to its fractional value drop below 1.0. Insufficient recognition of this value, and worse, the lack of action to control its value, is the main reason for low NUE both on the farm and worldwide [250,251].

The numerator of this equation is not the first but the second step in an effective control of the N_f_ use efficiency. The amount of N_f_ applied must meet at plant’s requirement for N to exploit its yield potential, taking into account both the stage of growth and the spatial variability in plant N status [252]. The effective determination of the amount of N_f_ requires the use of appropriate diagnostic tools. The first dose of N_f_, regardless of the crop, must be based on the content of N_min_ in the effective rooting depth of the currently cultivated plant [61]. A comprehensive view of the nutritional status of a plant in its full vegetation should be based on data on the content of both N_min_ and other nutrients in the soil [64]. The control of the plant nutritional status during the growing season, in fact, is limited to N. The chemometric diagnostic tools are good, but their use is limited. These methods are time-consuming and because of the delay between the sampling time and the delivery of the data to the farmer, they do not show the real condition of the plant N status. Real-time data can be obtained by using remote sensing techniques [253,254]. These methods rely on the absorption and reflection of solar radiation by a plant canopy. From this property of the plant, a number of crop characteristics can be determined in real time, such as (i) plant biomass, (ii) leaf area index, (iii) nitrogen content, (iv) chlorophyll content [255]. Biometric and nutritional data obtained at the cardinal stages of plant growth, combined with the required sum of the physiological effective temperatures at a given stage, form the basis for determining the crop growth rate. These data are used to forecast a plant’s demand for N in strictly defined stages of its development. This is the basis for determining the appropriate dose of N_f_. The spatial differences in the values of the field spectral indices can be used to develop a zonal map, showing the temporary crop N status. These maps are the basis for the application of N_f_ according to a plant’s requirements in a well-defined field area [36,256]. 

## 6. Conclusions

The production efficiency of all nutrients, applied as mineral fertilizers, can be evaluated mainly through their impact on nitrogen use efficiency (NUE). The effectiveness of nitrogen present in the soil/plant system depends on the degree of correction of soil factors limiting plant growth and nitrogen uptake at critical stages of yield formation by the currently cultivated plant by other fertilizers, including lime. There are a number of soil factors that limit nitrogen uptake and reduce NUE indices. Some of them can be easily controlled by the farmer, for example soil compaction, pH, organic matter as well as content of plant-available nutrients that improve metabolism and the use of N by plants. Moreover, regardless of the crop, an N dose must be based on the soil content of N_min_ in the effective rooting depth and/or plant nutrition status at critical growth stages. Improvement of the parameters characterizing FUE/NUE parameters can also be achieved through the proper selection and use of innovative fertilizers. In recent years, slow- and controlled-release fertilizers produced with the use of biochar, lignite or other carbon-containing organic compounds have been of particular interest. In addition to the standard advantages of this type of fertilizer, a positive effect on the physical and chemical properties of the soil, as well as the growth of the root system can be achieved. Nanofertilizers are a new, promising direction of fertilizer development. Of particular interest is the possibility of using fullarens as nutrients carriers. Unfortunately, a reliable assessment of nanofertilizers is limited by a relatively small amount of data from field trials. Summing up, it is worth noting that regardless of the solution used to improve the NUE indicators, each action has a positive effect on the biogeochemical cycle of biogenic elements, and at the same time can help to protect the environment and reduce fertilization costs.

## Figures and Tables

**Figure 1 plants-11-01855-f001:**
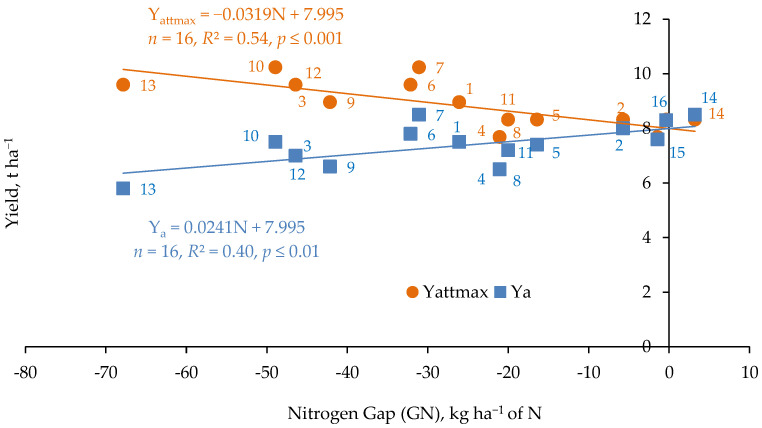
Diagram of yield trends in response to the nitrogen gap (NG) change. Example for winter wheat (based on Grzebisz and Łukowiak [8]). Key: Y_attmax_—maximum attainable yield; Y_a_—actual yield; 1–16 are the field numbers.

**Figure 2 plants-11-01855-f002:**
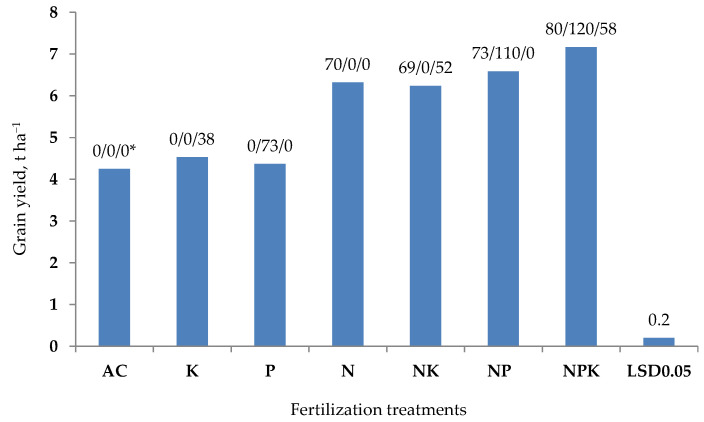
Effect of long-term differentiated fertilization on yield of winter wheat, mean of 2005–2008 years (own projection based on Blecharczyk et al. [17]). Key: AC—absolute control; K, P, N—experimental trials since 1957; LSD_0.05_—Least Significant Difference; 0/0/0*—respective values of nitrogen, phosphorus, and potassium use efficiency.

**Figure 3 plants-11-01855-f003:**
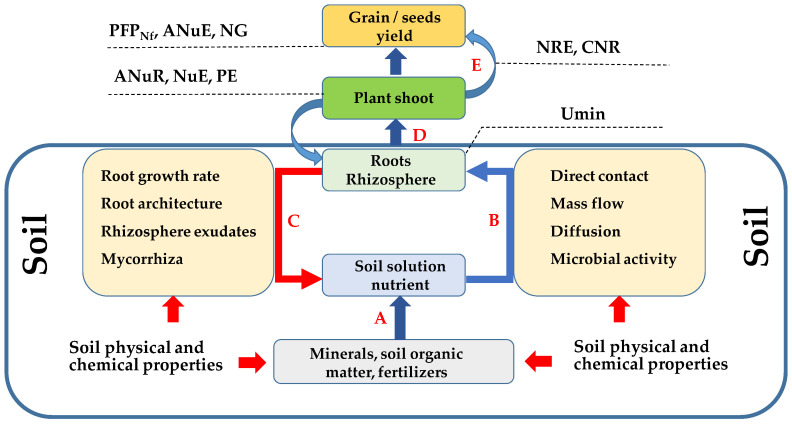
Fertilizer Use Effectiveness (FUE) indices in response to soil physical and chemical properties and processes responsible for nutrient uptake: (**A**) release of nutrients from solid phase; (**B**) processes of nutrient transport from the soil to the root surface; (**C**) the plant’s physiological response to conditions of nutrient supply; (**D**) processes of nutrient transportation to the plant shoot; (**E**) nutrient remobilization and transfer into grain/seeds. Blue arrows—transport processes; red arrows—influencing and feedback responses. FUE indices explanations: PFP_Nf_ —partial factor productivity of nitrogen; ANuE—apparent nutrient efficiency; NG—nitrogen gap; NRE—nitrogen remobilization efficiency; CNR—contribution of remobilized N to grain; ANuR—apparent nutrient recovery; NuE—nutrient uptake efficiency; PE—physiological N efficiency; U_min_—minimum uptake of a nutrient for the maximum rate of plant growth.

**Figure 4 plants-11-01855-f004:**
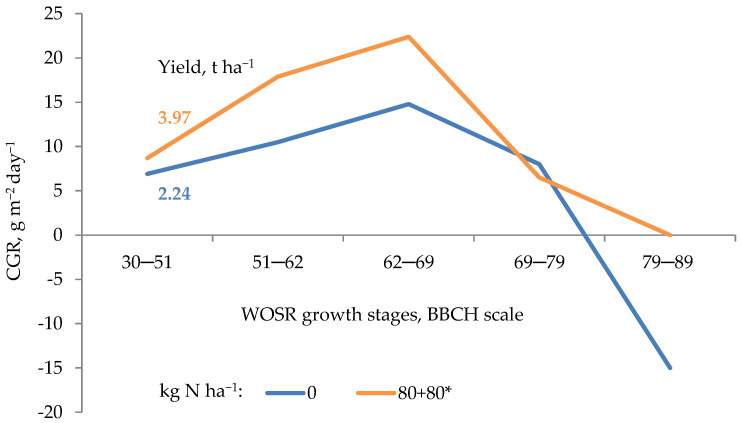
Crop growth rate (CGR) of winter oilseed rape (WOSR) during the growing season as affected by nitrogen fertilizer (based on Barłóg and Grzebisz [22]). Key: CGR—crop growth rate, N–0—absolute control, N–80 + 80*–N rate of 160 kg N ha^−1^ applied at the onset of the growing season restart in Spring; * ammonium nitrate; 30, 51, 62, 69, 79, 89—WOSR growth stages in BBCH scale.

**Figure 5 plants-11-01855-f005:**
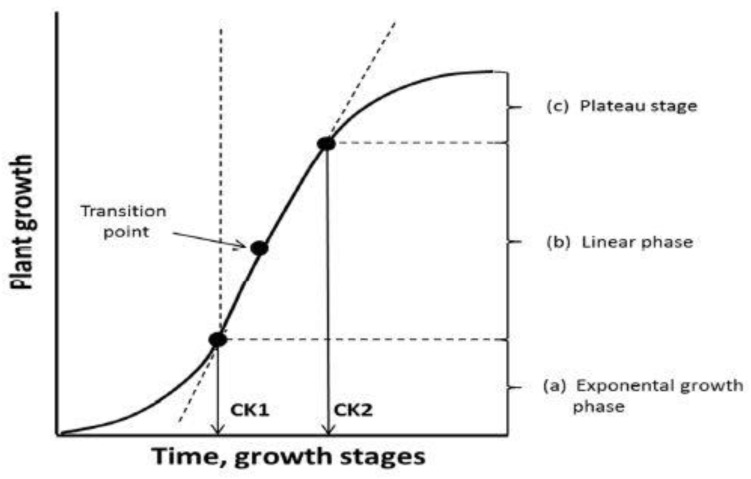
A conceptual pattern of dry matter accumulation by a typical seed/grain crop. Key: CK1, CK2—cardinal stage 1 and 2, respectively [8].

**Figure 6 plants-11-01855-f006:**
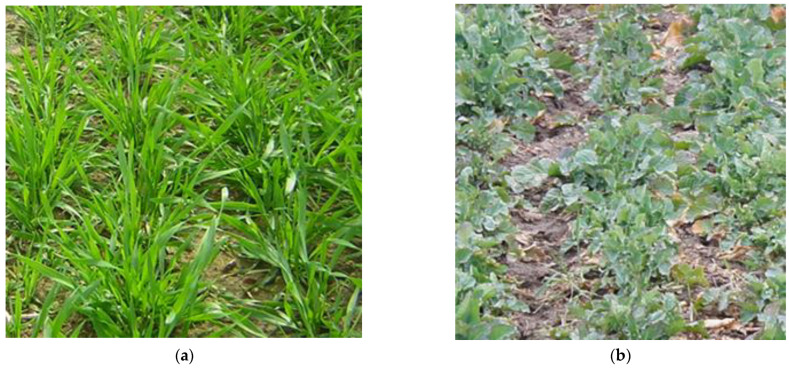
The Cardinal Stage 1 (CK1): winter wheat (monocot) (**a**) and winter oilseed rape (dicot) (**b**). Photos by W. Grzebisz.

**Figure 7 plants-11-01855-f007:**
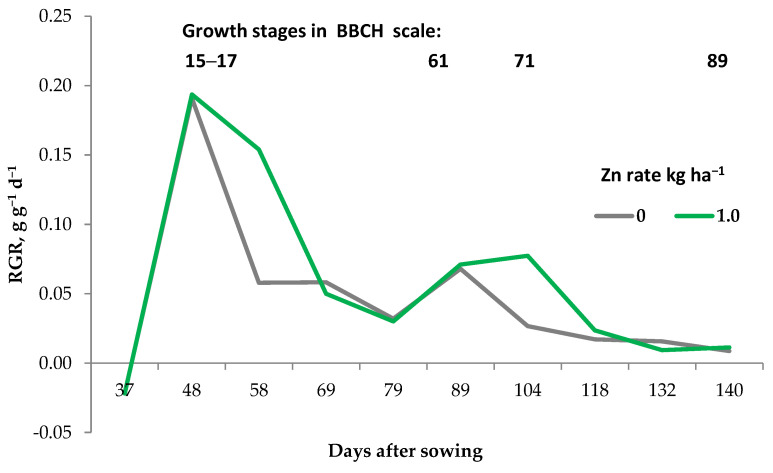
Relative growth rate (RGR) of maize during the growing season in response to foliar zinc (Zn) application (based on Grzebisz et al. [31]—modified).

**Figure 8 plants-11-01855-f008:**
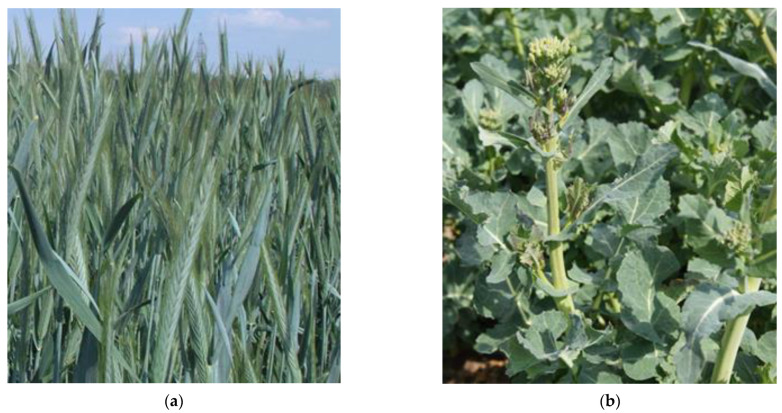
The Cardinal Stage 2 (CK2): winter rye (monocot) (**a**) and winter oilseed rape (dicot) (**b**). Photos by W. Grzebisz.

**Figure 9 plants-11-01855-f009:**
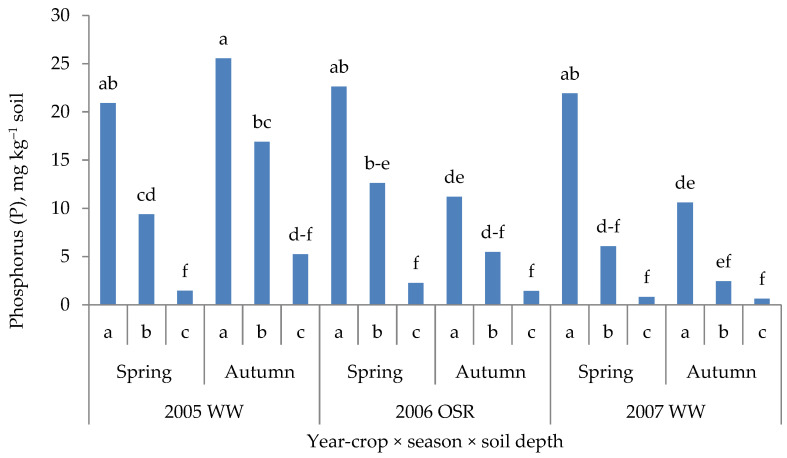
The seasonal patterns of available phosphorus distribution within soil layers (based on Łukowiak et al. [65]). Key: WW—winter wheat, OSR—oilseed rape. Letters indicate significant differences between treatments.

**Figure 10 plants-11-01855-f010:**
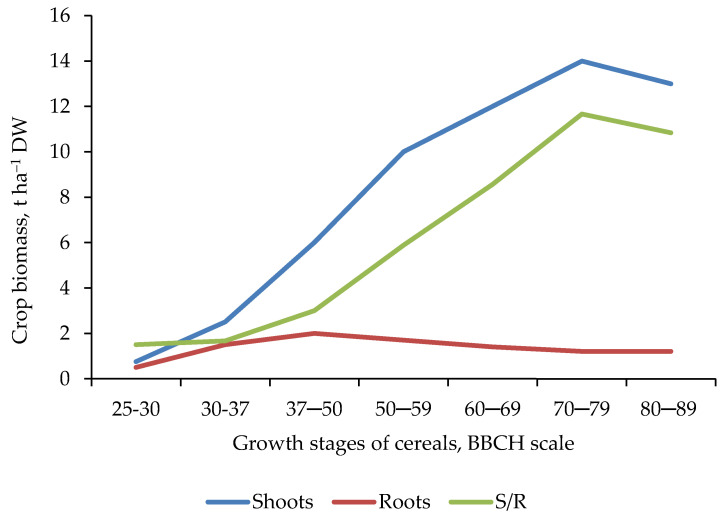
The general pattern of the growth of the root and shoot biomass of cereals during the growing season (based on Grzebisz [70]). Legend: R/S—root to shoot biomass ratio.

**Figure 11 plants-11-01855-f011:**
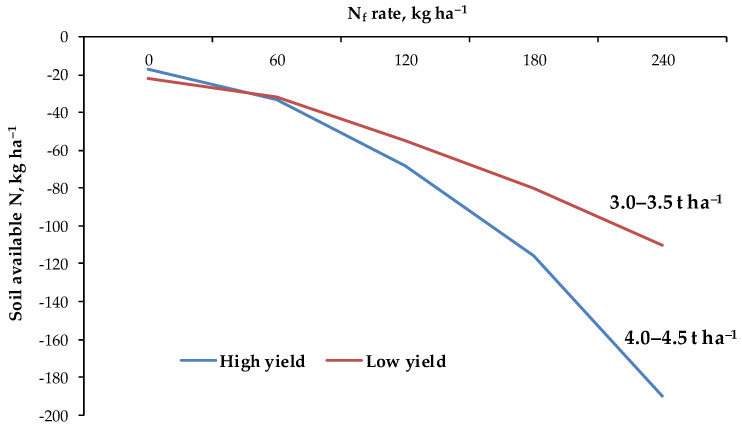
The amount of the soil and fertilizer N depleted during the Yield Formation Period (YFP) depending on nitrogen (N_f_) rate. Key: High, Low yield of winter oilseed rape. (Based on Grzebisz et al. [35]).

**Figure 12 plants-11-01855-f012:**
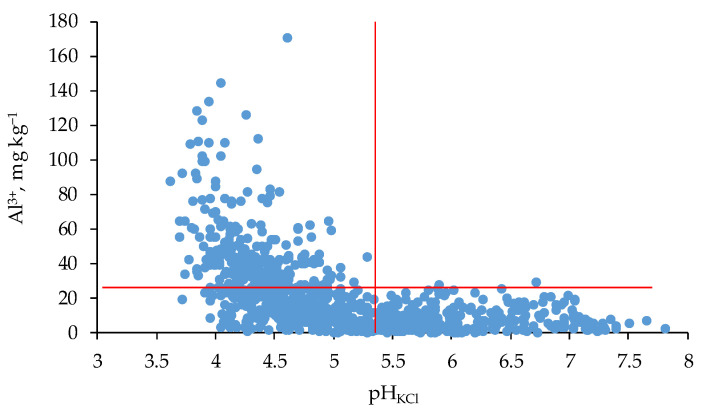
Exchangeable aluminum (Al^3+^) content as a function of soil pH measured in suspension of 1 molar KCl (1:2.5, *w*/*v*). Sandy soils, western Poland (*n* = 986). The red lines indicate the critical points for soil pH and Al^3+^ content. Source: Błaszyk [146].

**Figure 13 plants-11-01855-f013:**
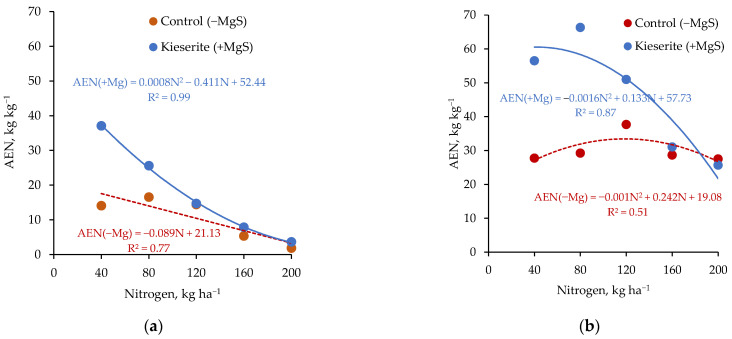
Effect of nitrogen application (40, 80, 120, 160 and 200 kg N ha^−1^) on the agronomic efficiency of nitrogen (AEN), calculated for white sugar yield of sugar beet, depending on the availability of magnesium in the soil—kieserite application at a rate of 24 kg Mg ha^−1^. Mean for two years for sandy soil (**a**) and loamy soil (**b**). Source: Pogłodziński et al. [182].

**Table 1 plants-11-01855-t001:** Sufficient ranges of key nutrient contents in crop plants at the first cardinal stage (CK1).

Nutrients (% or mg kg^−1^ of Dry Weight)	Maize		Sugar Beet	
	BBCH 17	BBCH 17	BBCH 41	BBCH 33
	Bergmann [28]	Schulte and Kelling [45]	Bergmann [28]	Barłóg [43]
Nitrogen, N, %	3.5–5.0	4.0–5.0	4.5–5.5	3.8–6.0
Phosphorus, P, %	0.35–0.6	0.4–0.6	0.3–0.6	0.27–0.46
Potassium, K, %	3.5–4.5	3.0–5.0	3.8–7.0	3.8–8.6
Magnesium, Mg, %	0.25–0.50	0.3–0.6	0.25–0.8	0.12–0.45
Calcium, Ca, %	0.3–1.0	0.51–1.6	0.6–1.5	0.28–0.85
Zinc, Zn, mg kg^−1^	30–70	25–60	20–80	15–45

**Table 2 plants-11-01855-t002:** Evaluation of maize nutritional status based on nutrient sufficiency ranges for the early leaf—the beginning of flowering—CK2.

Nutrients (% or mg kg^−1^ of Dry Weight)	Authors			
	Schulte and Kelling [45]	Jones et al. [46]	Campbell and Plank [41]	Potarzycki [33]
Nitrogen, N, %	3.0–4.0	2.6–3.6	2.8–4.0	2.1–3.33
Phosphorus, P, %	0.3–0.45	0.22–0.4	0.25–0.5	0.23–0.35
Potassium, K, %	2.0–3.0	1.8–4.5	1.8–3.0	1.9–2.5
Magnesium, Mg, %	0.2–0.8	0.43–1.0	0.25–0.8	0.41–0.67
Calcium, Ca, %	0.2–1.0	0.27–0.34	0.15–0.6	0.28–0.36
Zinc, mg kg^−1^	20–70	19–75	20–70	40–70 ^1^

^1^ corrected by author.

**Table 3 plants-11-01855-t003:** Coefficients of effective diffusion for main nutrients in water and soil solution ^1^.

Nutrient	Ion	D_w_, cm^2^ s^−1^	D_eff_, cm^2^ s^−1^
Nitrogen	NH_4_^+^	1.96 × 10^−5^	6.1 × 10^−8^
	NO_3_^−^	1.90 × 10^−5^	2.7 × 10^−6^
Phosphorus	H_2_PO_4_^−^	0.89 × 10^−5^	0.3–3.33 × 10^−9^
Potassium	K^+^	2.00 × 10^−5^	1–28 × 10^−8^

^1^ source: Raynaud and Leadley [76]; Clarkson [77].

## Data Availability

Not applicable.

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
