# Peer review of "Fertilizers and Fertilization Strategies Mitigating Soil Factors Constraining Efficiency of Nitrogen in Plant Production"

_plants, 2022, doi:10.3390/plants11141855_

Round 1

Reviewer 1 Report

The review paper is well written. I will suggest the authors the following:

1. check the English carefully once again

2. for me the title although good but confusing, better to revise carefully

3. the abstract do not show the significance of the paper, it is just generalizations better to report significant and new findings here. 

4. fertilizer research must have some recommendations for growers, there must be a paragraph for farmers usefulness

5. also report how to reduce the negative effects of N fertilizers in relation to Environment and higher costs

Author Response

Response to Reviewer 1 Comments

            Thank you for the many helpful suggestions to improve the manuscript. We have carefully and thoroughly revised the manuscript in response to the reviewer suggestions. In addition, the text has been corrected by Robert Kippen, a qualified native English speaking lecturer at the Adam Mickiewicz University in Poznań, Poland. We uploaded Word file with the Track Changes function. 

Answers to specific remarks and suggestions:

Point 1: check the English carefully once again

Response 1: the English was corrected once again by a native lecturer at the Adam Mickiewicz University in Poznań. The language correction is done in the Track Changes function.

Point 2: for me the title although good but confusing, better to revise carefully

Response 2: Thank you for the comment, the title has been corrected.

Point 3: the abstract do not show the significance of the paper, it is just generalizations better to report significant and new findings here. 

Response 3: Thank you for the comment, the abstract has been corrected. This applies to the middle part of the abstract.

Point 4: fertilizer research must have some recommendations for growers, there must be a paragraph for farmers usefulness

Response 4: Thank you very much for your valuable comment. The new version of the manuscript includes a new chapter 5. Among other things, it contains practical recommendations for farmers. Due to the new paragraph, new literature has also been added.

Point 5: also report how to reduce the negative effects of N fertilizers in relation to Environment and higher costs

Response 5: In our opinion, the new chapter gives instructions on how to reduce the negative effects of N fertilizers in relation to Environment and higher costs. Every effort to improve FUE / NUE indices not only is better use of nutrients, but also in reducing fertilization costs and negative impact on the environment. Therefore, the higher the values of NUE indices, the lower the risk of environmental pollution. This relationship is mainly related to the level of soil-plant interaction. The impact of fertilization on the environment should be treated more broadly than just the improvement of FUE / NUE. For example, the emission of ammonia during the storage of manure or the role of buffer zones in the migration of P into the environment should be discussed. However, this is a different area of consideration and requires a different article. In this article, we wanted to focus only on the possibilities of improving the nutrient use efficiency.

Reviewer 2 Report

I think the review is prepared in-depth. I am not sure that my revision should be useful, because I do not work with cereals…

However, I found this paper well-composed, and rich in references: it also offers a beginner´s guide of mechanisms that plants use to enhance the uptake of nutrients to the soil, mitigating soil factors constraining the efficiency of N.

I cannot judge if the present review is able to add relevant information to the several recent reviews published on this hot topic regarding FUE- fertilizer use efficiency, above all for nitrogen. However, in the concluding remark, I think that the authors should suggest what it should be done in the future for improving NUE and FUE.

It still needs a minor revision, following the points listed below, to be fully published. I suggest changing some English forms, although this language is not mine!  I’m sorry!

In the abstract, it is repeated two times ... necessary condition (lines 12 and 13). Please change.

Line 16; is not clear the sentence iii) neutralization of soil those soil factors that limiting growth of roots. Do you mean iii) neutralization of soil factors limiting the growth of roots?

Line 43: add “,” after (Nf)

Line 53 and 56: Please insert space and align (justify) the text

Line 113: show values much higher values

Line 118: in in

Line 206: please change Where: with where:

Line 223: I think that you should change identification with identifying

Line 274…and 490:  please cut (WOSR), because it has been defined previously

Line 276: Latest that N…. Is it correct? I think that should be: Latest N

Line 303 and 338: Please specific that values reported in table 1 and table 2 in  % or in mg/kg are expressed on dry weight …DW. 

Line 306…: You wrote “The biggest differences concern the content of Ca and K” Can you explain better why? Is it due only to the year effect or site effect? Similarly in table 2 I saw a great variability among sufficiency ranges reported. I think it is better to explain these variations.

Line 441: Developmenta

Line 474:  Please correct depend in depends

Line 479: of of

Line 626: Change N2 with N2

Line 740: Change In General…. In general

As I told you before, I think that should be very interesting to enlarge the conclusions, suggesting what it should be done in the future for improving NUE and FUE.

Best regards, Duilio Porro.

Author Response

Response to Reviewer 2 Comments

            Thank you for the many helpful suggestions to improve the manuscript. We have carefully and thoroughly revised the manuscript in response to the reviewer suggestions. In addition, the text has been corrected by Robert Kippen, a qualified native English speaking lecturer at the Adam Mickiewicz University in Poznań, Poland. We uploaded Word file with the Track Changes function.

Answers to specific remarks and suggestions:

I think the review is prepared in-depth. I am not sure that my revision should be useful, because I do not work with cereals…However, I found this paper well-composed, and rich in references: it also offers a beginner´s guide of mechanisms that plants use to enhance the uptake of nutrients to the soil, mitigating soil factors constraining the efficiency of N.

Point 1: I cannot judge if the present review is able to add relevant information to the several recent reviews published on this hot topic regarding FUE- fertilizer use efficiency, above all for nitrogen. However, in the concluding remark, I think that the authors should suggest what it should be done in the future for improving NUE and FUE.

Response 1: These suggestions were considered in the new version of the paper. 

 Point 2: It still needs a minor revision, following the points listed below, to be fully published. I suggest changing some English forms, although this language is not mine!  I’m sorry!

Response 2: These suggestions were considered in the new version of the paper.  The English was corrected once again by a native lecturer working at the Adam Mickiewicz University in Poznań. The language correction is done in the Track Changes function.

Point 3: In the abstract, it is repeated two times ... necessary condition (lines 12 and 13). Please change.

Response 3: Out of date, since the sentence was removed after considering the all suggestions of the Reviewer 1

Point 4: Line 16; is not clear the sentence iii) neutralization of soil those soil factors that limiting growth of roots. Do you mean iii) neutralization of soil factors limiting the growth of roots?

Response 4: Out of date, since the sentence was removed after considering the all suggestions of the Reviewer 1

Point 5: Line 43: add “,” after (Nf)

Response 5: The suggestion has been considered in the revised version.

Point 6: Line 53 and 56: Please insert space and align (justify) the text

Response 6: The suggestion has been considered in the revised version.

Point 7: Line 113: show values much higher values

Response 7: The text has been corrected

Point 8: Line 118: in in

Response 8: The text has been corrected

 Point 9: Line 206: please change Where: with where:

Response 9: The text has been corrected

 Point 10: Line 223: I think that you should change identification with identifying

Response 10: The suggestion has been considered in the revised version.

 Point 11: Line 274…and 490:  please cut (WOSR), because it has been defined previously

Response 11: The suggestion has been considered in the revised version.

Point 12: Line 276: Latest that N…. Is it correct? I think that should be: Latest N

Response 12: The suggestion has been considered in the revised version.

Point 13: Line 303 and 338: Please specific that values reported in table 1 and table 2 in  % or in mg/kg are expressed on dry weight …DW. 

Response 13: The suggestion has been considered in the revised version. The tables indicate that the nutrient contents are related to dry weight.

Point 14: Line 306…: You wrote “The biggest differences concern the content of Ca and K” Can you explain better why? Is it due only to the year effect or site effect? Similarly in table 2 I saw a great variability among sufficiency ranges reported. I think it is better to explain these variations.

Response 14: The suggestion has been considered in the revised version. The new version of the manuscript includes the sentences: “The biggest differences concern the content of Ca and K. The main reason for these variations is the calibration of plant tests under conditions of significant differences in the content of soil Ca and K in the area of the conducted research.”

Point 15: Line 441: Developmenta

Response 15: The figure 8 has been corrected

Point 16: Line 474:  Please correct depend in depends

Response 16: The text has been corrected

Point 17: Line 479: of of

Response 17: The text has been corrected

Point 18: Line 626: Change N2 with N2

Response 18: The text has been corrected

Point 19: Line 740: Change In General…. In general

As I told you before, I think that should be very interesting to enlarge the conclusions, suggesting what it should be done in the future for improving NUE and FUE.

Response 19: The suggestion has been considered in the revised version. The conclusion was corrected in line with the reviewer's remark